# TRAIL in the Treatment of Cancer: From Soluble Cytokine to Nanosystems

**DOI:** 10.3390/cancers14205125

**Published:** 2022-10-19

**Authors:** Hojjat Alizadeh Zeinabad, Eva Szegezdi

**Affiliations:** 1Apoptosis Research Centre, Biomedical Sciences Building, School of Biological and Chemical Sciences, University of Galway, H91 W2TY Galway, Ireland; 2Science Foundation Ireland (SFI) Centre for Research in Medical Devices (CÚRAM), Biomedical Sciences Building, University of Galway, H91 W2TY Galway, Ireland

**Keywords:** tumor necrosis factor-related apoptosis-inducing ligand (TRAIL), tumor-targeting, death receptor, nanoparticles, cancer, TRAIL resistance

## Abstract

**Simple Summary:**

TRAIL is a death ligand cytokine, predominantly used by effector immune cells to kill malignantly transformed cells. Since its discovery, TRAIL has attracted a lot of attention as a promising anticancer drug due to its selective action against cancer cells, promising a safe, low-toxicity treatment. Despite its promising characteristics, clinical trials have not delivered on this promise, due to issues with the poor in vivo biological activity of recombinant TRAIL formulations. Nanoparticles have the potential to overcome these limitations, and an increasing number of studies have reported very promising preclinical results. Here, we summarize the potential of TRAIL for cancer therapy, and provide a critical assessment of the challenges and the potential of various formulations of nanovehicles designed to date for TRAIL-based cancer therapy.

**Abstract:**

The death ligand tumor necrosis factor (TNF)-related apoptosis-inducing ligand (TRAIL), a member of the TNF cytokine superfamily, has long been recognized for its potential as a cancer therapeutic due to its low toxicity against normal cells. However, its translation into a therapeutic molecule has not been successful to date, due to its short in vivo half-life associated with insufficient tumor accumulation and resistance of tumor cells to TRAIL-induced killing. Nanotechnology has the capacity to offer solutions to these limitations. This review provides a perspective and a critical assessment of the most promising approaches to realize TRAIL’s potential as an anticancer therapeutic, including the development of fusion constructs, encapsulation, nanoparticle functionalization and tumor-targeting, and discusses the current challenges and future perspectives.

## 1. Introduction

The immune system plays an essential role in eliminating malignant cells by identifying and killing such cells. Death ligands (DL), expressed on the surface of effector immune cells, play a central role in this process. DLs are members of the tumor necrosis factor (TNF) cytokine family. They bind and activate death receptors (DR), thus inducing a regulated cell death program in the receptor-carrying cell.

The three best-known DLs are TNF itself, with its receptor, TNF receptor 1 (TNFR1), TNF-related apoptosis-inducing ligand (TRAIL), inducing cell death via the receptors DR4 and DR5, and Fas ligand (FasL) binding to the Fas receptor. Of the DLs, TRAIL received a lot of attention due to its capacity to induce apoptosis in many types of cancer cells without affecting healthy, non-transformed cells [1,2,3,4,5,6]. A variety of immune cells, including eosinophil granulocytes [7], macrophages [8,9], neutrophil granulocytes [10,11], dendritic cells [12,13], monocytes [9,14], natural killer (NK) cells [15], T and B cells [16,17], express TRAIL, and it is well established that TRAIL plays an important role in tumor immune surveillance, making TRAIL a promising candidate for an anticancer therapeutic [18,19,20].

Dulanermin, a soluble, recombinant human TRAIL (rhTRAIL) formulation comprising of the extracellular portion of TRAIL (amino acids 114–281) was the first TRAIL variant tested for the treatment of cancer [21,22]. The efficacy of Dulanermin either as a single agent or in combination with chemotherapy has been tested in several phase II and III clinical trials, but the tests have had disappointing results, showing very limited anti-tumor efficacy [4,23,24]. Although rhTRAIL had a very good safety profile and very low toxicity, it showed poor efficacy. The poor anti-tumor activity was linked to its short in vivo half-life, insufficient accumulation in tumor tissues and resistance of tumor cells to TRAIL-induced cell death [25,26,27].

To overcome these limitations, in the past ten years, novel, nanomedicine-based approaches have been considered. This review presents the most successful approaches used to improve the efficacy of TRAIL-based therapeutics highlighting how nanotechnology can help to realize the expected therapeutic potential of TRAIL and outlines the remaining challenges.

## 2. The TRAIL-Induced Apoptotic Pathway

The physiological role of apoptosis is to eliminate superfluous and abnormal cells, such as damaged or stressed cells including tumor cells [28]. It follows two major pathways; the intrinsic and the extrinsic apoptotic pathway (Figure 1) [29]. The intrinsic pathway, also known as the mitochondrial pathway, is initiated by events on the surface of mitochondria and regulated by the B cell leukemia 2 (Bcl-2) protein family [30,31]. The Bcl-2 family proteins are characterized by the presence of at least one of the four Bcl-2 homology (BH1–4) domains and divided into three sub-families [31,32]. The first sub-family consists of the founding member of the family, Bcl-2, BCL-2-like protein 1 (BCL2L1, Bcl-extra large (Bcl-x_L_), BCL-2-like-2 (Bcl-W), Myeloid cell leukemia-1 (Mcl-1), BCL-2-like-10 (Bcl-B) and BCL-2-related protein A1 (A1)). These members contain all four BH domains and they inhibit apoptosis, while the other two sub-families are pro-apoptotic. BCL2-related ovarian killer (Bok), Bcl-2 associated X protein (Bax) and Bcl-2 homologous antagonist/killer (Bak) contain the BH1-BH3 domains but lack the BH4 domain and they form the multidomain, pro-apoptotic sub-group of the family. Finally, the so-called BH3-only proteins form the third group and, as their name shows, they only contain the BH3 domain. Members of the BH3-only group include for example BCL-2 antagonist of cell death (Bad), BH3-interacting domain death agonist (Bid), BCL-2-like-11 (Bim), BCL-2-modifying factor (Bmf), BCL-2-interacting killer (Bik), Harakiri (Hrk, also known as death protein-5), Phorbol-12-myristate-13-acetate-induced protein 1 (Noxa) and BCL-2-binding component-3 (Puma). Upon stress, such as DNA damage, oxidative stress, endoplasmic reticulum stress, etc., BH3-only proteins are induced and/or activated and they initiate the apoptotic program by activating the multidomain pro-apoptotic Bcl-2 proteins, Bax and Bak [32,33,34,35,36], which then oligomerize and form pores on the mitochondrial outer membrane (MOM), resulting in its permeabilization [37,38]. From the permeabilized mitochondria, pro-apoptotic factors such as cytochrome *c* (Cyt *c*) [39], endonuclease G [40], apoptosis-inducing factor (AIF) [41] and second mitochondria-derived activator of caspase (Smac) [42] are released [43]. In the cytosol, Cyt *c* interacts with apoptotic protease-activating factor (Apaf-1) and pro-caspase-9 to form the apoptosome complex, which cleaves and thus activates pro-caspase-9, which in turn cleaves and activates the effector caspases, caspase-3/-6/-7, thus committing the cell to apoptosis [44,45]. The other pro-apoptotic factors released from the mitochondria facilitate this pathway. For example, Smac can neutralize X-linked inhibitor of apoptosis protein (XIAP), thus relieving XIAP-mediated caspase-3/-7/-9 inhibition [46,47].

The extrinsic apoptotic pathway is initiated by DRs (Figure 1). Binding of a DL to its DR results in the formation of a protein complex, called the death-inducing signaling complex (DISC), starting with the recruitment of the adaptor protein, Fas-associated death domain (FADD) and pro-caspase-8 and/or -10 to the death receptor. Pro-caspase-8 comprises of two N-terminal death effector domains (DEDs) followed by two protease catalytic domains (p18 and p10). One DED of pro-caspase-8 interacts with the same (DED) domain of FADD (homotypic interaction), while its second DED domain enables recruitment of an additional pro-caspase-8 molecule. Via the DED–DED interactions, multiple pro-caspase-8 molecules are recruited to the DISC, forming either a chain or a filament in which pro-caspase-8 activation takes place [48,49,50,51]. Once active, caspase-8/-10 are released from the DISC into the cytoplasm where similar to caspase-9, they activate downstream effector caspases, resulting in apoptosis (for review: Sessler et al., 2013; [52]).

In addition to effector caspases, caspase-8 can also cleave and activate the BH3-only protein, Bid, thus interconnecting the extrinsic and the intrinsic apoptotic pathways (Figure 1) [32,53]. Because in some cells DR-generated caspase-8 activity is not sufficient to activate the downstream caspase cascade, in these cells amplification of the apoptotic signal through the mitochondrial pathway (Bid activation) is essential to commit the cell to apoptosis (so-called type II cells) [54,55,56].

There are numerous mechanisms that control and override apoptotic signaling, thus allowing cells to survive. For example, the intrinsic apoptotic pathway is inhibited by the anti-apoptotic members of the Bcl-2 family which can block Bax/Bak activation and consequent Cyt *c* release [19]. This is a common mechanism of DL-resistance in type II cells, and often employed by cancer cells [57,58,59,60].

More specific to the extrinsic apoptotic pathway, activation of DRs themselves is controlled by decoy receptors, such as decoy receptor (DcR)1, DcR2 and osteoprotegerin (OPG) for TRAIL, or DcR3 in the case of FasL [61,62,63]. Downstream of the DRs, pro-caspase-8 activation can be inhibited by cellular FLICE inhibitory protein (cFLIP), a pseudo-caspase that interacts with and prevents pro-caspase-8 activation in the DISC [48,49,64]. Further downstream, the apoptotic pathway is controlled by inhibitor of apoptosis (IAP) proteins, including XIAP (X-linked IAP), cIAP1 (cellular IAP1) and cIAP2 [46,65,66]. XIAP can directly bind and inhibit caspase-3, caspase-7 and caspase-9 [66,67,68], while cIAP1 and cIAP2 inhibit apoptosis through activation of the nuclear factor-κB (NF-κB) signaling pathway upon DR-activation leading to the induction of cFLIP [69], Bcl-2 and Bcl-x_L_ [69,70,71,72,73].

## 3. TRAIL Formulations to Increase Its Serum Half-Life Time

Soluble TRAIL has a short in vivo half-life time of 0.56–1.02 h [21] that limits its efficacy [74]. The main reason for it is its relatively small size of approximately 60 kDa, in its biologically active, trimeric form. As kidney filtration cut-off is around 70 kDa [75], many proteins smaller than this size, including rhTRAIL, are quickly cleared from the blood. To overcome this issue, numerous recombinant TRAIL derivatives have been designed to increase its size by fusing it to a peptide or protein [76,77,78,79,80,81,82,83,84,85,86,87]. For example, TRAIL has been conjugated to serum albumin, either directly [88] or indirectly [89], which increases its circulatory half-life substantially. Additionally, because albumin can interact with albondin (gp60 receptor) on endothelial cells, the fusion also facilitated the transport of TRAIL across the endothelium into the tissues via caveolae-mediated transcytosis [90] leading to enhanced tumor-suppressing potential [88].

A similar strategy has been used by Brin and colleagues [91]. They fused arginine deiminase to rhTRAIL, as arginine deaminase was known to up-regulate DR5 expression. They showed that the fusion construct was more effective against a colorectal cancer tumor xenograft than rhTRAIL.

All these studies could achieve substantial increase in the circulatory half-life of TRAIL; however, it would be important to determine whether these modifications alter or influence receptor binding, as the fusion partners were quite large and thus, they could alter the conformation of TRAIL’s receptor-binding motifs.

## 4. TRAIL Formulations to Mimic Membrane-Bound TRAIL

Although the soluble form of TRAIL, comprising of its extracellular domain, is biologically active, it has been shown that the full-length, membrane-bound form of TRAIL is a 100–1000-fold more potent inducer of cell death ascribed to an oligomeric or clustered conformation of membrane-bound TRAIL as opposed to the trimeric structure of soluble TRAIL [92,93]. To address this issue, a variety of nanocarriers, including organic and inorganic nanoparticles (NPs), viral NPs and cells or cellular components have been assessed to replicate the oligomeric TRAIL complexes believed to be formed in the membrane upon receptor binding.

### 4.1. Cell- and Virus-Based Delivery Tools

One approach to mimic the structure and organization of membrane-bound TRAIL was the use of viral particles. For example, TRAIL has been conjugated to capsid protein IX of the Ad5 oncolytic adenovirus via its intracellular, N-terminus using a heterodimeric zipper domain thus enabling an orientation of TRAIL on the virus surface similar to that of on the plasma membrane [94]. The study found that this TRAIL formulation had increased cytotoxic potential both in vitro and in vivo. In a recent work by the same lab [95] this design was further improved by combining the oncolytic virus treatment with ginsenoside-Rh2 treatment. Ginsenoside-Rh2 is a derivative of the small molecule drug, dammarane found in ginseng species and has been shown to up-regulate TRAIL receptor expression [96] through which the study could achieve higher toxicity in tumor cells with low baseline TRAIL receptor expression, such as primary AML cells.

The rod-shaped flexuous filamentous potato virus X (PVX) is another virus tested for TRAIL delivery [97]. PVX is a unique, multifunctional vehicle increasingly considered for nanomedicine, because the surface of PVX is rich in cysteine and lysine side chains, which can be easily used as adaptors for functionalization [98,99,100]. His-tagged TRAIL could be immobilized on the virus surface via His-tag-nickel-nitrilotriacetic acid (Ni-NTA) interaction. Placing the His-tag on the N-terminus of TRAIL enabled conjugation of TRAIL to the surface of the virus in its native orientation. This formulation led to a 3–10-fold higher pro-apoptotic potential in vitro compared to rh-His-TRAIL. Furthermore, it also suppressed in vivo tumor growth in a human triple-negative breast cancer mouse model [97].

Several studies explored whether human cells can be used as TRAIL-delivery tools. Using cells as a vehicle for TRAIL therapy has several advantages, such as low immunogenicity [101], ease of genetic-engineering [102,103] and tumor-targeting using migratory properties of the cell vehicles (e.g., mesenchymal stem cells (MSCs) [103] or via other genetic modifications) [104]. In a smart approach [105], genetically engineered platelets expressing surface-bound TRAIL have been developed using a lentiviral transgene expression. Hematopoietic stem and progenitor cells (HSPCs) were transduced with a TRAIL gene whose expression was under the megakaryocyte-specific integrin αIIβ promoter. By transplanting the TRAIL-transduced HSPCs into the bone marrow of mice, TRAIL-expressing platelets were produced in vivo, during megakaryocyte maturation. This design reduced prostate cancer metastasis, indicating that TRAIL-expressing platelets could eliminate circulating tumor cells [105]. In a similar study, Shah and colleagues [106] engineered neural stem cells to secrete soluble TRAIL. This approach markedly decreased tumor burden and prolonged survival in mouse models of glioblastoma [106].

Another important cell type tested as TRAIL-delivery vehicles are mesenchymal stem cells (MSCs) due to their ability to migrate and home to sites of injury [107,108]. Based on this hypothesis, several studies have shown that TRAIL-expressing MSCs significantly inhibited tumor growth and prolonged survival in various mouse cancer models, including lung cancer [109], lung metastasis of breast cancer [110] and renal cell carcinoma [111], glioblastoma [102,112,113,114], hepatocellular carcinoma [115], cervical carcinoma [116], colorectal carcinoma [117] and pancreatic carcinoma [118].

Despite the advantages of cell-based TRAIL therapy, cell therapy also has considerable limitations, including the limited replicative potential of MSCs [119], the requirement of GMP (good manufacturing practice) facilities for generation of cellular therapeutics and the cost and time required to obtain sufficient number of cells for therapy and potential immunogenicity [120,121]. To overcome these problems, nanoparticle carriers are being developed that can not only mimic cell-based TRAIL therapy [122], but also provide a mechanism to combine several advantageous properties of TRAIL delivery from increased half-life, correct conformation, combination with other drugs to targeted delivery to tumor sites [123,124,125,126,127].

### 4.2. Nanoparticle-Based TRAIL Delivery

NPs have a great advantage over biological carriers, similar to cells and EVs, due to their high flexibility, which allows selective targeting of the tumor site facilitating cargo accumulation in the tumor microenvironment and thus enhancing efficacy, while also reducing potential toxic effects on normal cells. An additional benefit of nanoparticles is the possibility to encapsulate chemotherapeutic agents and co-deliver them with TRAIL, therefore overcoming the frequent issue of tumor TRAIL resistance.

The delivery of TRAIL via NPs has been extensively explored, and several successful nanoparticle-based TRAIL-delivery systems have been developed (Figure 2). These studies also confirmed that NPs can enable TRAIL oligomerization, increase serum half-life and overcome tumor TRAIL resistance, particularly when the TRAIL-NPs were combined with sensitizing agents [128,129,130,131,132].

A variety of nanocarriers, including metallic NPs [134,135,136], polymeric NPs [137,138,139,140], lipid-based NPs [141,142,143], protein NPs [128,144,145,146] and carbon-based NPs [132,147], have been tested as TRAIL carriers. Key findings of these studies are summarized in Figure 3.

In the following, we focus on discussing the most successful nano-based approaches for TRAIL delivery.

#### 4.2.1. Cell Component-Based Nanoparticles for TRAIL Delivery

Since 2011, when particle-coating with cell membrane was first reported [148], a variety of cell types have been used as membrane sources to coat nanoparticles [149] and thus combine the advantages of designed nanoparticles with membrane properties of a specific cell type to enable tissue targeting, NP immune escape or prolonged blood circulation [150,151,152]. For example, monodispersed silica particles were coated with membranes of activated platelets followed by functionalization with TRAIL with the aim to target and kill circulating tumor cells (CTCs). In the circulation, tumor cells are vulnerable and to protect themselves from immune recognition, they hide from effector immune cells by binding and covering themselves with platelets (cloaking/microthrombi). The study showed that the propensity of cloaking can be turned against CTCs as CTCs bound the platelet membrane-coated NPs instead of platelets, which exposed them to TRAIL and killed them [153].

Exosomes are another cell component being explored as drug delivery tools for cancer therapy [154,155,156]. Exosomes were first described in 1981 [157] as small extracellular vehicles (EVs) of 30–100 nm diameter surrounded by a lipid bilayer, naturally released by various cell types [158]. Due to their native origin, they have low immunogenicity matched with an excellent ability to carry a wide spectrum of cargoes ranging from hydrophilic as well as hydrophobic drugs to genetic materials, making EVs promising nanocarriers [154,159].

Protein-based delivery tools: TRAIL protein conjugated to the surface of various NPs (e.g., liposomes, viruses, carbon nanotubes, metallic NPs) interacts with DR4/DR5 on the tumor cells to induce apoptosis. Some TRAIL protein-based delivery tools are functionalized with targeting ligands, e.g., antibodies, or molecules (i.e., hyaluronic acid) that preferentially target tumor cells and enable the induction of cell death via TRAIL. The unique structure of liposomes makes them an excellent delivery tool for combination therapy that can co-deliver chemotherapeutic reagents and TRAIL protein resulting in a synergistic cell death signal. Metallic NPs functionalized with TRAIL protein not only induce apoptosis via TRAIL, but they can also generate reactive oxygen species (ROS) as they can be internalized into tumor cells. The generated ROS leads to the release of pro-apoptotic proteins from mitochondria and activation of the intrinsic apoptosis pathway, thus amplifying the TRAIL death signal.

Tumors are also known to release EVs [160] containing various cytokines [161,162], immunosuppressive and inhibitory proteins such as transforming growth factor beta 1 (TGFβ1) [163], programmed death ligand 1 (PD-L1) [164,165,166], cyclooxygenase-2 (COX2), cytotoxic T-lymphocyte-associated protein 4 (CTLA4), TRAIL [167] and FasL [168,169]. Using the ability of cells to produce and release EVs, it was demonstrated that cells genetically modified to ectopically express TRAIL released exosomes decorated with biologically active TRAIL [170]. As an example, the Huber laboratory transduced K562 leukemic cells with TRAIL to produce TRAIL-expressing exosomes. These exosomes induced apoptosis of melanoma and lymphoma cells in vitro and reduced tumor growth in vivo in melanoma and lymphoma mouse models [170]. Similarly, Shamili and colleagues [171] reported that exosomes isolated from cultures of MSCs engineered to ectopically express TRAIL could induce apoptosis in vitro as well in vivo in a melanoma mouse model [171]. Another study [172] demonstrated that TRAIL-containing MSC-derived exosomes could induce cell death in a range of cancer cell lines including cell lines resistant to soluble TRAIL [172].

Despite all the above-mentioned unique advantages, EVs are highly heterogeneous [173] and their isolation and purification is challenging, time-consuming and expensive in comparison to artificially generated vesicles, such as liposomes [174,175,176], which drove the development of liposome-based TRAIL nanovehicles.

#### 4.2.2. Liposome-Based TRAIL Delivery

So far, liposomes are one of the most commonly used NPs for cancer therapy with the first formulations in clinic use already, such as the sustained-release formulation of cytarabine (DepoCyt^®^) [177] or pegylated liposomal doxorubicin (DOXIL^®^/CAELYX^®^) [178].

Since 1961, liposomes have been the most widely investigated cargo delivery system due to good biocompatibility [179,180,181], biodegradability, ability to encapsulate both hydrophilic and hydrophobic cargos, a wide range of sizes and monodispersed composition. Easy surface modification and high liposome permeability offered solutions to cargo targeting and controlled release, which earned liposomes the title of “smart drug carriers” [176,182]. Furthermore, the high dynamic mobility of liposome membranes makes their structure similar to biological membranes and a good choice for artificial cell generation [183]. Although liposomes are not as stable as exosomes, they can be easily synthesized with the desired size on a large scale. Moreover, surface modification and functionalization with peptides or proteins, including antibodies, is very easy [176,184]. A final notable advantage of liposomes that makes them unique is the possibility to externally control the release of the encapsulated cargo at the tumor site [185,186].

One of the first studies developing TRAIL-functionalized liposomes was the Martinz-Lostao laboratory [143], who used large unilamellar vesicles (LUV) in which Ni^2+^ cations in the LUVs bound His-tagged TRAIL. This noncovalent interaction bound high amounts of TRAIL without changing the structure of the liposome and maintaining TRAIL bioactivity. The study proved that immobilization of TRAIL to liposomes increased TRAIL’s local concentration leading to increased pro-apoptotic potential compared to rhTRAIL. Additionally, LUV-TRAIL was also efficient against rhTRAIL-resistant leukemia cells [143], likely due to the arrangement of TRAIL into highly ordered oligomers on the LUVs which could induce a high level of DR4/DR5 clustering on the leukemic cells leading to increased pro-caspase-8 activation able to drive a type I extrinsic cell death pathway [129,187].

Due to the hyperpermeability of the vasculature in tumor tissues, the incorporation of TRAIL in nanoparticles can facilitate its preferential accumulation in tumor sites, a process called passive targeting [188] (Table 1). On the contrary, as a soluble protein, rhTRAIL can also pass through normal vessels, leading to a non-selective distribution of TRAIL between tumor sites and healthy tissues, thus lowering its concentration in the tumor.

Although the majority of FDA-approved nanoparticles for cancer therapy have used passive targeting [192,193], it has been shown that the decoration of nanoparticles with recognition molecules, such as an antibody to target tumor-specific antigens, enhances efficacy [194,195,196,197]. This is also true for TRAIL, with the first studies showing that active targeting of TRAIL to tumor sites can further increase its in vivo efficacy (Table 2).

#### 4.2.3. Active Targeting

One of the most significant advantages of nanoparticles in cancer therapy is the ability to functionalize their surface with proteins and other biomolecules to selectively target tumor cells [198,199]. The most common strategies for NP tumor-targeting are (1) targeting NPs to immune cells and using the immune cells as NP carriers to indirectly target tumor cells, (2) functionalization with monoclonal antibodies to directly target unique or overexpressed antigens on the surface of tumor cells, and (3) functionalization with ligands/receptors against tumor-expressed receptors/ligands, such as hyaluronic acid (HA) to target CD44, often highly expressed on cancer cells [200,201].

Of these three strategies, indirect targeting of tumor cells is probably the most widely used. In this method, NPs are linked to immune cells, and the immune cells deliver the NPs to the tumor site or to CTCs. An appealing example of indirect targeting was the study by Mitchell and colleagues, who developed liposomes functionalized with E-selectin and TRAIL via Ni-NTA binding [202]. E-selectin enabled the liposomes to bind to E-selectin ligands widely expressed by leukocytes [203,204] as well as CTCs and to eliminate the CTCs via TRAIL-induced apoptosis thus preventing prostate cancer metastasis [205]. In a similar study, tethering a very low dose of TRAIL and E-selectin to liposomes was sufficient to induce TRAIL-mediated cell death in circulating tumor cells, inhibit metastasis and prolong survival in the 4T1 breast carcinoma mouse model [206].

NK cells have also been used as a carrier of TRAIL-functionalized liposomes. Functionalizing liposomes with TRAIL and anti-NK1.1 antibodies allowed the liposomes to bind to NK cells ubiquitously expressing NK1.1 to form “super-NK cells”. The super-NK cells displayed high cytotoxicity both in vitro and in vivo compared to liposomes functionalized with TRAIL alone or to unmodified NK cells [207,208].

Targeting NPs to tumor cells via immune cell-mediated delivery relies heavily on the ability of the immune cells to find the tumor cells. As tumors evolve, they acquire abilities to inhibit immune cells or to hide from them by removing antigens immune cells would recognize (the process of immune escape) [209,210], which is a major limitation of indirect, immune cell-based NP targeting and nanosystems with the ability to target the tumor directly may be more effective. For example, Seifert and colleagues [211] engineered liposomes able to target epidermal growth factor receptor (EGFR, overexpressed on a variety of cancers) [212,213] by functionalizing the liposomes with an EGFR-specific single-chain Fv molecule (EGFR scFv) and TRAIL (immune-LipoTRAIL) [211]. The immune-LipoTRAIL NP did not only increase TRAIL’s serum half-life, but it could target and inhibit the growth of an EGFR-positive colorectal tumor xenograft. Although, in this study, EGFR targeting did not enhance the efficacy of the TRAIL-liposome, direct NP targeting of tumors warrants further research [211].

**Table 2 cancers-14-05125-t002:** Active targeting approaches for TRAIL delivery.

Formulation	TRAIL Form/Location	Targeting Strategy	Tumor Type	Ref.
Inhalable HSA NPs	rhTRAIL/Surface	HSA to target gp60 transcytosis pathway	Lung cancer, mouse model	[214]
HSA NPs	rhTRAIL/Surface	Transferrin on HSA NPs to target transferrin receptor on tumor cells and HSA to target gp60 transcytosis pathway	Colorectal cancer, mouse model	[215]
Liposome	pTRAIL/Surface	Angiopeptide-2 to target the low-density LRP on BBB and glioma cells	Glioblastoma, mouse model	[216]
Polymeric NPs coated with platelet membrane	rhTRAIL/Surface	Platelet membrane to target CD44 on tumor cells via *p*-selectin	Metastatic breast cancer, mouse model	[217]
Polymeric NPs	pTRAIL/Inside	Reconstituted HDL on polymeric NPs to target scavenger receptor class B type I on MSCs	Pulmonary melanoma, mouse metastasis model	[218]
Polymeric NPs	pTRAIL/Surface	EGFR-specific peptide to target EGFR on laryngeal cancer cells	Hep-2 laryngeal squamous cell carcinoma, mouse model	[219]
PEI-coated gold nanocomposite	pTRAIL/Surface	Dexamethasone to target nucleus	Hep3B cell, mouse model	[220]
Polymeric NPs	rhTRAIL conjugated to PEG/Inside	HA to target CD44 on tumor cells	Collagen-induced arthritis, mouse model	[221]
Liposome inside a hyaluronic acid crosslinked-gel shell	rhTRAIL/ B/w liposome and gel shell	HA to target CD44 on tumor cells	MDA-MB-231, breast cancer xenograft model	[222]
ZnFe_2_O_4_ magnetic NPs w/mesoporous silica shell with PEI	TRAIL DNA/Surface	Adipose tissue-derived MSCs treated with NPs to target tumor cells	Ovarian cancer, mouse model	[223]
Carbon dot coated polyethyleneimine	pTRAIL/ Surface	MSCs for targeting tumor cells	Lung cancer cell line (A549 cells)	[224]
PEG-crosslinked albumin hydrogel	rhTRAIL/Inside	HSA to target gp60 transcytosis pathway	Pancreatic cancer, mouse model	[225]
Magnetic ternary nanohybrids (iron oxide NPs coated with HA)	TRAIL DNA/Inside	MSCs for targeting tumor cells	Glioma, mouse model	[226]
Polymeric NPs	pTRAIL/Inside	Choline-derivate to target choline transporters on BBB and glioma cells	Glioma, mouse model	[227]

Abbreviations: HSA, human serum albumin; NPs, nanoparticles; rhTRAIL, recombinant human TRAIL; pTRAIL, plasmid TRAIL; LRP, lipoprotein receptor-related protein; BBB, blood-brain barrier; HDL, high-density lipoprotein; MSCs, mesenchymal stem cells; EGFR, epidermal growth factor receptor; PEI, polyethylenimine; PEG, polyethylene glycol.

#### 4.2.4. Nanoparticles for TRAIL-Based Drug Combination Therapies

Although tethering TRAIL to cells or NPs significantly improved its therapeutic efficacy, primary tumors tend to be resistant to TRAIL. TRAIL resistance develops during tumor pathogenesis, as a mechanism to escape from immune recognition and elimination [74,228,229,230,231,232,233,234]. Accordingly, approximately two thirds of cancer cell lines have been found to be TRAIL-resistant, emphasizing that TRAIL resistance is a major limitation of TRAIL-based therapeutics [235].

Combination therapies have become the primary strategy to address tumor drug resistance, for which nanotechnology offers an excellent tool (Table 3) due to its capacity to simultaneously deliver multiple therapeutic reagents [236,237].

#### 4.2.5. Combination of Chemotherapeutic Reagent Encapsulating-Nanoparticles with TRAIL

Since numerous studies revealed the synergistic cytotoxic effect of TRAIL with chemotherapeutic drugs [244], the unique potential of nanoparticles to encapsulate drugs may provide an opportunity to overcome TRAIL resistance. In this regard, liposomes and polymeric NPs received the most attention because of their advantageous properties. Incorporation of peptides and drugs in polymers has been known since the 1950s, with the first polymeric nanoparticles reported in 1976 [188]. Since then, various synthetic and biological polymers such as polyethylene glycol (PEG), polyglutamic acid, N-(2-hydroxypropyl) methacrylamide (HPMA), polylactic acid (PLA) and poly(lactic-co-glycolic acid) (PLGA) have been developed and been used in the clinic [245]. The key advantages offered by polymeric nanoparticles are their ability (1) to encapsulate a variety of therapeutic molecules such as RNA, DNA, proteins and chemotherapeutic drugs, (2) for controlled cargo-release, and (3) for surface functionalization with ligands and antibodies [246,247,248].

The first studies aimed to co-deliver a sensitizing agent used already approved chemotherapeutics known to induce DR4/DR5 expression by triggering cellular stress thus sensitizing cancer cells to TRAIL, such as derivatives of the microtubule inhibitor, taxol and DNA damaging agents, such as doxorubicin and mitoxantrone [240,249]. These NP formulations were more effective than the agents alone, paving the way for the development of more complex TRAIL-drug combination nanoformulations incorporating active tumor cell targeting and enhancing anti-tumor immune response.

Dendrimer, a polymer NP type, was a choice of carrier for many of these studies. Dendrimers are highly branched, globular polymers with internal cavities. This structure allows them to easily conjugate and encapsulate various drugs, proteins, and other molecules. In addition, they are biocompatible [250,251]. Liu and colleagues engineered a pH-sensitive dendrimer to co-deliver DOX and a TRAIL-encoding plasmid (pORF-hTRAIL) for glioma treatment [252]. They conjugated DOX and a T7 peptide that targets brain tumor cells by binding to the transferrin receptor (TfR), often overexpressed in glioblastoma [253] and brain capillary endothelial cells [254,255] to a poly-l-lysine dendrimer loaded with the pORF-hTRAIL plasmid. When the T7 peptide bound to the TfR on the tumor cells, the NP became endocytosed. Acidic pH in the endosomes then induced the release DOX and pORF-hTRAIL from the pH-sensitive dendrimers, thus simultaneously delivering DOX that triggered DNA damage and the TRAIL gene, which, after production of TRAIL protein, induced apoptotic signaling in glioma cells [252]. Several other dendrimer formulations, including PAMAM [256], diaminotriazine-modified PAMAM [189] and fluorinated dendrimers [257], have been used to increase the cytotoxicity of TRAIL by co-delivering a TRAIL-sensitizing drug and a TRAIL-expression cassette.

Another approach targeting NP delivery to tumor cells used an acrylamide-based doxorubicin nanogel (DOX-NG) fabricated by the single emulsion method [217]. For targeting, the DOX-NGs were coated with platelet membrane since platelets express *p*-selectin [258], a molecule that binds to CD44, a receptor often up-regulated on cancer cells [259]. Finally, the coated DOX-NPs were functionalized with TRAIL using cysteine thiol-conjugation. This nanoplatform decreased the number of circulating tumor cells and lung metastases in vivo and significantly reduced the number of tumor metastases [217]. In a similar approach, Suryaprakash and colleagues designed a hybrid TRAIL-MSC/nanocomposite spheroid system to target glioblastoma (GBM) cells [260]. The nanocomposite encapsulated mitoxantrone (MTX, a genotoxic drug/topoisomerase II inhibitor) and its surface was functionalized with a peptide to target interleukin-13 receptor alpha 2 (IL13Rα2), which is frequently overexpressed in GBM cells [261]. Finally, a spheroid hybrid TRAIL-MSC/nanocomposite was also shown to be effective [260]. These were generated by passing TRAIL-secreting MSCs and modified nanocomposites through a microfluidic system. This hybrid spheroid had the advantage of expressing TRAIL in a true, membrane-bound conformation.

Feng and colleagues addressed the question of whether NP-based TRAIL delivery could instigate an anti-tumor immune response, thus facilitating long-term disease control using another polymer NP, periodic mesoporous organosilica (PMO) [262]. DOX was encapsulated into a PMO NP and the NP surface was modified with TRAIL. This nanoplatform induced an immune response by activating dendritic cells, CD4^+^ and CD8^+^ T cells and inhibited tumor growth in a breast cancer mouse model [262].

More recent studies used liposomes to deliver TRAIL-chemotherapeutic drug combinations. This nanoplatform resulted in remarkable cell death compared to single treatments with DOX or liposome-presented TRAIL-NPs in the H-1080 melanoma mouse model [263]. In a similar study, TRAIL-functionalized liposomes loaded with polymeric micelles modified with piperlongumine, a natural alkaloid that triggers oxidative stress also showed enhanced TRAIL-induced apoptosis in TRAIL-resistant prostate cancer cell lines [264].

Overall, these data show that nanotools that can co-deliver TRAIL and chemotherapeutic agents significantly enhance the efficacy of TRAIL, making these nanoplatforms a promising therapeutic tool as they can overcome tumor TRAIL resistance by simultaneously activating both the intrinsic and the extrinsic apoptosis pathways.

#### 4.2.6. Combination of ROS Producing-Nanoparticles with TRAIL

Metal NPs have been reported to induce cell death through inducing oxidative stress [265,266] owing to the fact that they could generate reactive oxygen species, such as singlet oxygens (^1^O_2_) [267,268,269,270,271] or hydrogen peroxide (H_2_O_2_) [272]. In the presence of ferrous or cuprous ions, H_2_O_2_ can be converted into OH^•^ through the Fenton reaction [269,273]. OH^•^ then reacts with proteins, lipids and DNA that damages the cell and leads to cell death [274]. Production of OH• from H_2_O_2_ is not limited to the ferrous or cuprous ions, other ions such as Fe, Mn, Cu, Co, etc., also can generate OH^•^ through Fenton-like reactions [272,275,276]. Since oxidative stress has been shown to sensitize some cancer cells to TRAIL, numerous studies have demonstrated that co-delivering TRAIL with metal NPs could achieve the same effect [184,277]. The challenge in using metal NPs is their limited flexibility for oriented conjugation of proteins, such as TRAIL and accordingly, studies to date have predominantly relied on electrostatic- and amide-based conjugation on iron and silver NPs (Figure 3) [278,279].

Although the delivery of TRAIL via these NPs is a promising approach, it can be challenging to link proteins to metal NPs in an oriented and stable manner. This can lead to the unspecific distribution of the metal NPs throughout the body causing widespread oxidative stress linked to kidney toxicity [280]. ROS can also disrupt cell membranes, including that of healthy cells, emphasizing that the safe application of metal NPs would require their selective targeting to cancer cells [281].

#### 4.2.7. Combination Phototherapy with TRAIL

Recently, NP-based phototherapies such as photothermal therapy (PTT) and photodynamic therapy (PDT) have received considerable attention for cancer treatment. NP-based phototherapies can directly induce cell death through generating ROS (in PDT) and/or converting light energy into heat (in PTT) [282,283,284,285] and the first studies assessing this methodology to overcome TRAIL resistance have produced promising results. They encapsulated iron oxide nanoparticles and photothermal metallo-aromatic complexes (Ph556) into a hybrid micellar NP to generate a nanocomposite, and the surface of the nanocomposites was decorated with TRAIL via electrostatic interactions (TRAIL nanocomposites). They found that TRAIL nanocomposites significantly improved efficacy against TRAIL-resistant lung tumor xenografts (A549 cells) and the sensitization was associated with DR4/DR5 induction [286]. In another approach, outer membrane vesicles (OMV) from *E. coli* ectopically expressing human TRAIL were functionalized with an ανβ3 integrin-targeting ligand and the photothermal agent, indocyanine green [287]. Upon NIR irritation, indocyanine green produced singlet oxygen that led to oxidative stress and consequent TRAIL sensitization of resistant tumors [287]. Based on these results, TRAIL can be incorporated into PTT nanosystems without a problem of losing its biological activity. These systems enable not only the active targeting of tumor cells based on tumor antigens, but also site-specific activation of multiple pro-apoptotic mechanisms to effectively eradicate tumor cells while posing minimal risk to non-malignant tissues. To date, the design and generation of these nano-approaches is however labor-intensive and needs further optimization.

## 5. Conclusions and Future Perspectives

One of the most promising mechanisms to selectively eliminate malignant cells is by inducing their suicide program using death ligands. Regarding the death ligands, TRAIL has received the most attention as a potential cancer therapeutic molecule as it induces apoptosis selectively in cancer cells without affecting healthy, non-malignant cells. However, the low biological activity of TRAIL as a soluble protein, its short in vivo half-life time, insufficient accumulation in tumor tissues and acquired resistance of tumor cells to TRAIL-induced cell death showed poor efficacy in clinical trials.

In the last two decades, nanotechnology has opened a new avenue for cancer therapy, including that for TRAIL. Conjugating TRAIL to nanoparticles has addressed the poor pro-apoptotic activity of soluble rhTRAIL and prolonged its circulatory half-life. Furthermore, co-encapsulating drugs and conjugating TRAIL to NPs with particular properties, such as metal-based NPs, has led to overcoming TRAIL resistance in specific cancer models. Additionally, the co-functionalization of NPs with TRAIL and monoclonal antibodies or other molecules such as hyaluronic acid, has allowed specific tumor-targeting, thus enhancing TRAIL accumulation in tumor sites.

Overall, TRAIL-based NPs could address most of the limitations of soluble rhTRAIL therapy, and these nanoformulations have the potential to become a potent and safe treatment for cancer. However, all these formulations are still in preclinical development stage, and none of them have been translated to the clinic yet. The following considerations may pave the way to the design of an effective TRAIL-based nano-system that can potentially translate into the clinic.

-Most TRAIL-NP formulations have only been tested in immunodeficient mouse models engrafted with cancer cell lines. These models cannot replicate crucial properties of the human body to assess targeting efficiency, the heterogeneity of primary tumors or the role of the immune system. Testing in patient-derived tumor xenografts or genetically engineered mouse models is crucial to progress TRAIL-NPs towards clinical development.-With the advance of personalized therapy, it is becoming possible to target the exact molecular mechanisms of TRAIL resistance in individual tumors. Cells become sensitive to TRAIL during the first stages of malignant transformation [288], but this sensitivity is lost as the tumor evolves by the up-regulation of one (but not multiple) inhibitory pathways (e.g., increased Bcl-2 expression, XIAP expression) [289]. On the contrary, healthy cells have redundant (multiple) mechanisms to maintain their TRAIL resistance [289]. Targeting the specific apoptosis inhibitor, instead of combining TRAIL with a broad-spectrum, and often toxic cell stressor (e.g., doxorubicin), is likely to achieve higher efficiency and/or lower systemic toxicity.-In most of the current formulations for co-delivery of TRAIL and TRAIL sensitizers, TRAIL is conjugated to the surface of the NPs, and its sensitizers are loaded inside the NPs. This means that first, TRAIL engages with DRs on the target cell, and after that, or in parallel, when the cell endocytoses the NPs, the sensitizing agent can exert its effect, when it might be too late. Formulations for time-dependent release nanoparticles, or development of sequential treatment regimes, might solve this issue.-Selective tumor-targeting with higher-specificity tumor markers have a great potential to maximize safety as well as potency of TRAIL-NPs and enable controlled activation and/or cargo-release. Thus, identifying and targeting more specific cancer biomarkers, such as CLL-1 (targeting leukemic stem cells) is required in order to minimize off-target toxicity [290].

Overall, nanoparticles have opened a vast array of opportunities and mechanisms to overcome the limitations of the traditional, soluble protein-based TRAIL therapy, and they also offer a spectrum of solutions to personalize TRAIL-based cancer therapy. The results of multiple preclinical models prove that in the right formulation TRAIL has a strong therapeutic potential which warrants progression into more advanced preclinical models and towards clinical trials.

## Figures and Tables

**Figure 1 cancers-14-05125-f001:**
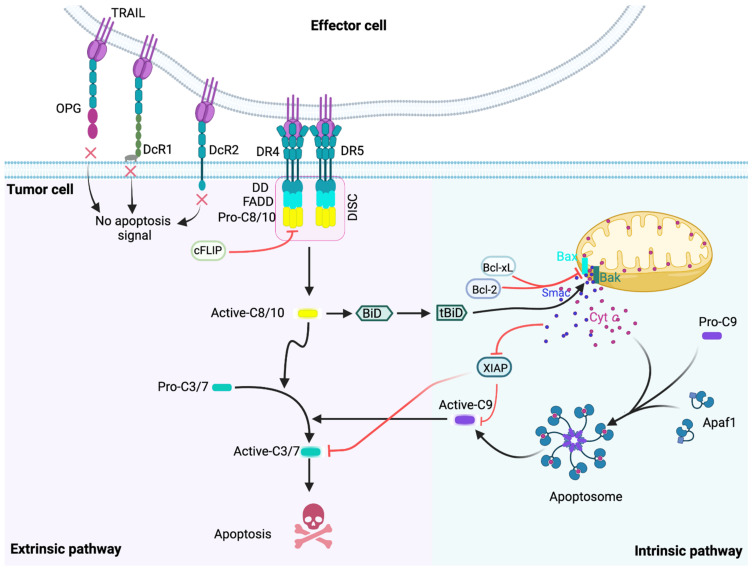
The TRAIL-induced cell death pathway and its regulation. Interaction of TRAIL with DR4 and DR5 leads to the recruitment of the adaptor protein, FADD, which then binds pro-caspase-8 and/or -10 leading to their activation. Activated caspase-8/-10 are released from the DISC into the cytoplasm, where they activate the downstream, effector caspases, which by cleaving vital proteins commit the cell to die. Caspase-8 can also cleave the BH3-only protein, Bid thus activating it. Truncated Bid (tBid) then binds and activates Bax and Bak, leading to mitochondrial membrane permeabilization and release of apoptotic factors such as Cyt *c* and Smac into the cytosol. Cyt *c* binds to APAF-1 thus initiating its oligomerization and recruitment of pro-caspase-9. The formed protein complex is called the apoptosome, serving as an activation platform for pro-caspase-9. Activated caspase-9 cleaves and activates the effector caspases, caspase-3/-6/-7, thus amplifying the DR-initiated apoptotic signal. Apoptosis signaling is kept under control at all stages of the process. First, decoy receptors (DcR1, DcR2) and osteoprotegerin (OPG) can sequester TRAIL thus preventing DR4/5 activation. cFLIP can interact and inhibit pro-caspase-8 in the DISC. At the mitochondrion, antiapoptotic Bcl-2 proteins, e.g., Bcl-2 itself and Bcl-x_L_ inhibit Smac and Cyt *c* release. XIAP can inhibit caspase-3/-7/-9 activation; while Smac can block XIAP, thus allowing the apoptotic program to progress. TRAIL, tumor necrosis factor (TNF)-related apoptosis-inducing ligand; DR, death receptor; OPG, osteoprotegerin; DcR, decoy receptor; DD, death domain, FADD, Fas-associated death domain; Active/Pro-C, pro-caspase; DISC, death-inducing signaling complex; cFLIP, Cellular FLICE inhibitory protein; Cyt *c*, cytochrome *c*; APAF-1, apoptotic protease-activating factor-1; Smac, XIAP, X-linked inhibitor of apoptosis protein; Bak, BCL-2 antagonist/killer; Bax, Bcl-2 associated X protein. Figure was generated with BioRender.

**Figure 2 cancers-14-05125-f002:**
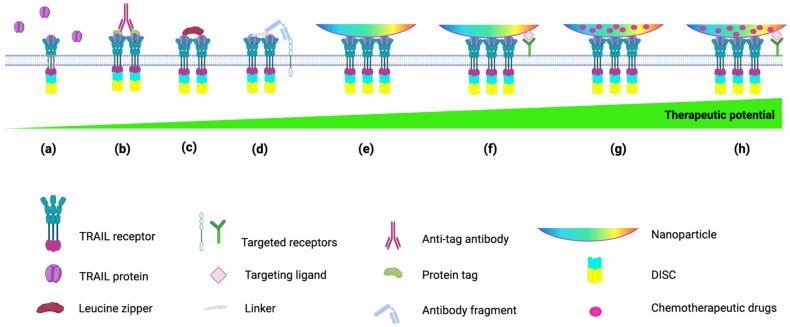
Correlation between TRAIL formulations and anticancer therapeutic potential. (**a**) RhTRAIL-based therapy, (**b**) crosslinking TRAIL monomers through an anti-tag antibody or (**c**) protein fusions facilitating TRAIL oligomerization (i.e., leucine zipper-TRAIL) [5,133]. (**d**) Conjugation of antibody fragment to TRAIL using a linker (i.e., scFv425) [86] to target tumor cells. (**e**) Conjugation of TRAIL to nanoparticles (NPs) to mimic membrane-bound TRAIL, (**f**) additional functionalization of the NPs with antibodies or other molecules to target cancer cells or tumor sites. (**g**) Encapsulation of chemotherapeutic agents into TRAIL-functionalized NPs and (**h**) further modification of NPs surface with molecules or antibodies for active targeting. Figure was generated with BioRender.

**Figure 3 cancers-14-05125-f003:**
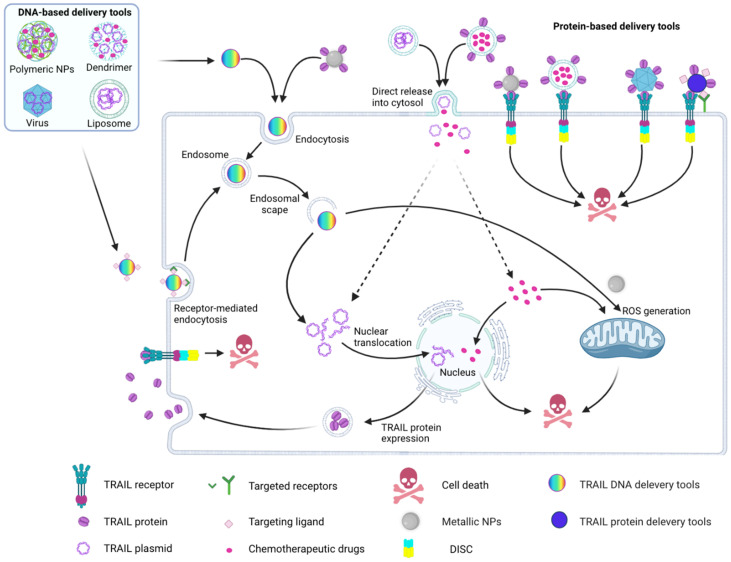
Schematic representation of using nanoparticles to improve TRAIL efficacy. TRAIL DNA-based delivery tools: Uptake by tumor cells via either passive- or active (receptor-mediated) endocytosis. The endocytosed NPs escape from the endosome and release the encapsulated TRAIL gene into the cytosol, which then translocates to the nucleus. After transcription and translation, soluble TRAIL is released into the tumor microenvironment, where it can interact with its receptors (death receptor (DR)4 and 5 to induce apoptosis. Alternatively, TRAIL gene-encapsulating liposomes can fuse with the cell membrane and directly release the TRAIL gene into the cytosol. Dendrimers and polymeric NPs can co-deliver TRAIL gene and chemotherapeutic reagents, resulting in DNA damage thus amplifying the TRAIL-induced cell death signals. The figure was generated with BioRender.

**Table 1 cancers-14-05125-t001:** Nanoformulations used for passive TRAIL delivery.

Formulation	Size (nm)	TRAIL Form/Localization	Main Findings	Tumor Type	Ref.
Liposome	100–140	rhTRAIL/Surface	Mimicking the membrane properties of natural TRAIL increased receptor clustering and improved cytotoxic potential of TRAIL thus overcoming resistance to soluble TRAIL.	Colorectal cancer, mouse model	[141]
Liposome	100	rhTRAIL/Surface	Enhanced clustering of DR5 overcoming resistance to soluble TRAIL.	Colorectal cancer, mouse model	[129]
Triazine modified dendrimer	200	pTRAIL/Surface	Enhancement in transfection efficacy of TRAIL gene. Improved tumor growth inhibition.	Osteosarcoma mouse model	[189]
PEG-TRAIL microencapsulated into PLGA	11,000–15,500	TRAIL/Inside	Increased biological half-life time and sustained TRAIL delivery up to 18 days.	Colorectal cancer, mouse model	[190]
Single-walled carbon nanotubes	ND	rhTRAIL/Surface	20-fold increase in TRAIL cytotoxicity against cancer cells without toxicity against normal cells.	Colorectal cancer, NSCLC, hepatocellular carcinoma	[191]

Abbreviations: nm, nanometer; rhTRAIL, recombinant human TRAIL; pTRAIL, plasmid TRAIL; DR, death receptor; PEG, polyethylene glycol; PLGA, poly (lactic-co-glycolic acid); ND, not determined; NSCLC: non-small cell lung carcinoma.

**Table 3 cancers-14-05125-t003:** Nanoparticles for combinatorial approaches for TRAIL delivery.

Formulation	TRAIL Form/Location	Strategy to Overcome TRAIL Resistance	Tumor Type	Ref.
Inhalable HSA NPs; loaded w/Dox	rhTRAIL/Surface	DNA damage caused by DOX.	Lung cancer, mouse model	[214]
HSA NPs; loaded w/DOX	rhTRAIL/Surface	DNA damage caused by DOX.	Colorectal cancer, mouse model	[215]
Liposome, loaded w/DOX	rhTRAIL/Surface	DNA damage caused by DOX.	NSCLC, mouse model	[238]
Liposome, loaded w/PTX	pTRAIL/Surface	PTX induced M-phase cell cycle arrest.	Glioblastoma, mouse model	[216]
Polymeric NPs coated with platelet membrane, loaded w/DOX	rhTRAIL/Surface	DNA damage caused by DOX.	Metastatic breast cancer, mouse model	[217]
Polymeric NPs, loaded w/DOX	rhTRAIL/Inside	DNA damage caused by DOX.	Prostate and colon cancer, mouse model	[239]
PEI-coated gold nanocomposite	pTRAIL/Surface	ROS generation by iron oxide NPs inducing DNA damage.	Hep3B cell xenograft mouse model	[220]
Inhalable highly porous PLGA microparticles, loaded w/DOX	rhTRAIL/Surface	DNA damage caused by DOX.	Mouse model of H226 cell metastasis	[240]
Liposome inside a hyaluronic acid crosslinked-gel shell, loaded w/DOX	rhTRAIL/Between liposome and the gel shell	DNA damage caused by DOX.	MDA-MB-231 breast cancer cell, xenograft mouse model	[222]
Chitosan modified magnetic nanoparticles	pTRAIL/Inside	Hyperthermia induced by a magnetic field.	Pulmonary metastatic mouse model	[241]
Alginate modified CaCO_3_ NPs, loaded w/DOX	rhTRAIL/Surface	DNA damage caused by DOX.	Cervical cancer cell line (HeLa cells)	[242]
Magnetic ferric oxide NP	rhTRAIL/Surface	ROS generation by iron oxide causing DNA damage.	Glioma mouse model	[243]
Polymeric NPs, DOX intercalated	pTRAIL/Inside	DNA damage caused by DOX.	Glioma mouse model	[227]

Abbreviations: HSA, human serum albumin; NPs, nanoparticles; rhTRAIL, recombinant human TRAIL; DOX, doxorubicin; DNA, deoxyribonucleic acid; NSCLC, non-small cell lung cancer; pTRAIL, plasmid encoding TRAIL; PTX, paclitaxel; PEI, polyethylenimine; ROS, reactive oxygen species; PLGA, poly (lactic-co-glycolic acid).

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
