# Peer review of "TRAIL in the Treatment of Cancer: From Soluble Cytokine to Nanosystems"

_cancers, 2022, doi:10.3390/cancers14205125_

Round 1
Reviewer 1 Report
It is a well-written review that successfully summarizes the current bibliography concerning the TRAIL’s potential as anticancer therapeutics within the framework of nanoparticle functionalization. There is a limited number of reports in this particular scientific area, thus it is a significant contribution to this field. Some minor modifications are needed, though:
Section 4 and its subsections provide a large amount of information and the generation of Figure 2&3 and Table 1 are not sufficient to include and summarize all the important knowledge. Authors must create at least 2 additional Tables (for 4.2.1 – 4.2.7) in order to be more reader friendly.
Authors must check the manuscript for syntax errors like double spaces, etc
Author Response
Comment 1: Section 4 and its subsections provide a large amount of information and the generation of Figure 2&3 and Table 1 are not sufficient to include and summarize all the important knowledge. Authors must create at least 2 additional Tables (for 4.2.1 – 4.2.7) in order to be more reader friendly.
Response: As the reviewer requested, we have generated 3 separate tables each summarising one type of nanoformulation to present the discussed studies in a more structured and transparent manner.
Comment 2: Authors must check the manuscript for syntax errors like double spaces, etc
Response: We thank the reviewer spotting these typographical errors, which have now been corrected in the manuscript.
Reviewer 2 Report
This review summarizes the current literature reports about TRAIL-associated monotherapies as strategies to concur the limitation of TRAIL-mediated anti-cancer effect. Overall, the review covers most of the updated studies and gives a future perspective on the potential TRAIL-mediated treatment. Some minor revisions are needed.
Are there current clinical trials undergoing to evaluate nontechnique-mediated delivery of TRAIL therapy? If some, they can be summarized in this review.
Is the Figure on Page 2 a graphic figure? Some abbreviations such as rh are suggested to list full names.
The citation locations of each sentence should be included before the sentence close period.
Some locations have additional spaces such as line 40, The efficacy…
Conflicts of Interest should be included at the end of manuscript.
Author Response
Comment 1: Are there current clinical trials undergoing to evaluate nontechnique-mediated delivery of TRAIL therapy? If some, they can be summarized in this review.
Response: We agree with the reviewers that providing information about clinical trials undergoing of nontechnique-mediated delivery of TRAIL is important. In section 5, the most important TRAIL clinical trials are mentioned. Unfortunately, to date none of the nano-based TRAIL delivery systems have been translated to the clinic. We provided some possible reasons that why these formulations remained in preclinical stage in the section 5 in the manuscript.
Comment 2: Is the Figure on Page 2 a graphic figure? Some abbreviations such as rh are suggested to list full names.
Response: We thank the reviewer to point this out. Yes, Figure on page 2 is a graphical abstract and now we added the abbreviations to the figure legend.
Comment 3: The citation locations of each sentence should be included before the sentence close period.
Response: We thank the reviewer pointing out this error. It is now corrected throughout the manuscript.
Comment 4: Some locations have additional spaces such as line 40, The efficacy…
Response: We thank the reviewer spotting these typographical errors, which have now been corrected in the manuscript.
Comment 5: Conflicts of Interest should be included at the end of the manuscript.
Response: As requested we included it now.